# Translating Data Science Results into Precision Oncology Decisions: A Mini Review

**DOI:** 10.3390/jcm12020438

**Published:** 2023-01-05

**Authors:** Enrico Capobianco, Marco Dominietto

**Affiliations:** 1The Jackson Laboratory, 10 Discovery Drive, Farmington, CT 06032, USA; 2Paul Scherrer Institut, Forschungsstrasse 111, 5232 Villigen, Switzerland

**Keywords:** data science, radiomics, decision systems

## Abstract

While reviewing and discussing the potential of data science in oncology, we emphasize medical imaging and radiomics as the leading contextual frameworks to measure the impacts of Artificial Intelligence (AI) and Machine Learning (ML) developments. We envision some domains and research directions in which radiomics should become more significant in view of current barriers and limitations.

## 1. Introduction

### 1.1. Starting Point

Data Science is de facto inspiring a new generation of integrative inference models justified by the emergence of diversity, heterogeneity, and complexity in data structures required to support reproducible and generalizable results. The related issue of scalability, one that comes with big data, is well managed by leveraging artificial intelligence (AI) and machine learning (ML).

What has become critical is combining data-driven knowledge with applicable clinical expertise directly into AI/ML tools. This would mean optimizing the translational power and emphasizing the interpretability and explainability of the complex model results. The bottleneck here refers to the role of validation in determining the significance of the results, given the impossibility of systematically resolving complexities at both structural data and computational levels.

In oncology, for example, it is relevant for clinical decisions to assess any data- or model-driven insight in terms of measurable impacts. However, these ultimately depend on established and constantly optimized workflows linking diagnosis, treatment, and follow-up. Therefore, only indirect impacts may be found.

### 1.2. Diagnosis

Cancer classification is based primarily on standardized staging and relies on a large amount of anatomical and physiological information acquired from medical imaging devices, histology, blood sample biomarkers, genetic profiling, etc. Repositories aimed at creating public cancer data collections, such as NCI’s Genomic Data Commons integrating resources such as The Cancer Genome Atlas (TCGA), provide solutions for storing, sharing, and analyzing data interactively within the research community (see https://ccr.cancer.gov/, and https://www.cancer.gov/about-nci/organization/ccg/research/structural-genomics/tcga, both accessed on 29 December 2022).

Naturally enough, an accurate classification drastically improves the possible personalization of the treatment and increases the probability of efficacy. However, diagnostic procedures create large amounts of data that are normally underused in clinical practice. Such data sets include patient populations and healthy subjects, in part informing on screening procedures and acting as control groups in specific studies. The patient group is more stratified according to tumor staging or pathological clinical assessment.

### 1.3. Treatment

Treating cancer implies a combination of targeted approaches, such as surgery and radiation therapy, together with systemic treatment via drug delivery, as with chemotherapy. Data-driven resources have become available, such as the Therapeutically Applicable Research to Generate Effective Therapies (TARGET) (https://ocg.cancer.gov/programs/target, accessed on 29 December 2022). In general, the goal of treatment personalization requires a sequence of tasks adapted to patient-specific characteristics and following clinical protocols.

The data complexity is a variable that largely depends on the type of treatment. Therefore, big data in this area is complex but keeps evolving [1]. As the data repositories mature, they will incorporate more shared tools for data analysis and require more sophisticated quality metrics.

For instance, radiation therapy releases a precise amount of ionizing radiation dose to the tumor region conservatively, i.e., to save surrounding healthy tissues. Dose optimization is done according to the patient’s anatomy and can only be determined by using high-resolution (millimetric) 3D medical images. Then, the choice of the best surgical procedure depends on gathering accurate anatomical and physiological information to allow complete tumor resection and minimize the damage to healthy organs [2]. Systemic treatments, either chemo- or immune therapy, rely on physiology and histology data.

In general, treatment-related data complexity is multifaceted and not anchored to specific characteristics such as spatial distribution (typical of anatomy) but rather to the ability to perform high-dimensional capture of most of the possible interactions among the various pathological/physiological processes for which measurements are obtained [3].

### 1.4. Follow-Up

Temporally scheduled follow-up is used to constantly monitor the progression or regression of the disease. Follow-up tasks generate simpler data types aimed at checking signs or symptoms indicating the risk of relapse or the side effects of treatment. In general, a patient’s examination is required to ascertain whether it is either disease-free or progression (recurrence or metastasis). From a prognostic standpoint, follow-up is crucial because it allows retrospective data analysis and ultimately addresses through survival the success/failure of treatment while annotating patient records about key clinical factors like local tumor control and development of side effects [4].

## 2. Trials

Oncological research based on clinical trials and screening programs is usually supported by multi-centric studies (cost-effective, resource-intensive, rigor/quality preservation). With reference to possible experiments aimed at determining new treatment effects or the performance of devices or procedures applied to patients, clinical trials include a number of phases [5] covering the behavior of new drugs through dose ranges and estimates of toxicity over a quite limited number of patients, the assessment of drugs biological activity on larger groups of subjects, then the consideration of efficacy tests versus standard therapies and side effects over a significant cohort of patients (typically in the order of few thousands), and finally commercialization.

Large clinical trials are normally public, and their data and results can be found online (see the European Medicine Agency https://www.clinicaltrialsregister.eu, and the National Library of Medicine https://clinicaltrials.gov, both accessed on 29 December 2022). The data sets originated from clinical trials are quite heterogeneous and depend on the types of experiments. Data from Phase 0 and phase 1 consist of a large amount of information normally confined to a small number of patients. In fact, the goal is to accurately characterize drug pharmaco-dynamics and side effects by measuring all the possible physiological modifications in terms of blood sample values, DNA damages, metabolic changes, anatomical modification, etc. On the contrary, phases 3 and 4 deal with larger patient cohorts and produce a larger amount of patient data but with a smaller amount of information (the rationale being to test the hypothesis previously formulated and minimize time and costs). A comprehensive list of databases available online to the scientific community is reported in [6]. For image data, the results from the clinical trial are publicly available from repositories like the Cancer Imaging Archive (TCIA) (https://www.cancerimagingarchive.net, accessed on 29 December 2022).

## 3. Precision Oncology

### 3.1. Background

Big data analytics developments are driving cancer research along two mainstream paths. One looks at the prediction of tumor behavior and the response to therapies through the analysis of large heterogeneous patient data sets. This path operates by adopting new computational strategies that model all factors underlying the variability among subjects. The other path looks at personalized therapies and follows the paradigm of precision medicine [7,8]. The goal is to collect patient-specific information to guide the decisions about the best possible treatment. Radiation therapy offers a good example as the delivered dose distribution is decided according to patient-specific anatomy [9]. Similarly, systemic treatments such as chemo- or immune therapies are administered according to the histology and physiology of individuals.

Focusing on a single patient seems vaguely in agreement with large data set analysis. However, there are different aspects of tumor heterogeneity [10] to be considered. At one end, inter-subject heterogeneity reflects the variability of tumor behavior among patients, which can be optimally modeled when considering a large population of subjects. This is because increasing the number of subjects makes it possible to better stratify for tumors with similar characteristics, thus improving the diagnostic resolution.

At another end, there is an intrinsic heterogeneity in each tumor, i.e., the intra-subject variability [11], depending on the features specific to angiogenesis, proliferation, necrosis, hypoxia, pH values, etc. The same therapy may have different effects on patients depending on such characteristics. For example, radiation therapy will have a strong effect on oxygenated tissues but will only partially affect hypoxic areas [12]. The reason is that radiation-induced effects are mediated by oxygen and are proportional to its concentration. This already offers a rationale for acquiring more information on each patient to customize treatment [13].

### 3.2. Big Data from Single Patient

An in-depth multidimensional study of each patient is needed to increase the chances of successful treatment. Together with molecular profiling of tumor cells that informs on patient-specific genetic alterations and, in principle, allows the development of targeted drugs for the identified altered proteins, personalized treatment can rely on additional information [14] compared to genetic markers, i.e., physiology and anatomical properties of both tumors and surrounded tissues, disease familiarity, personal lifestyle, environmental conditions, etc. Elucidating this multilayered data architecture is the challenge that big data presents to address two main questions: (a) How can patient-specific data be used to predict treatment outcomes? (b) What method is justified, and why is preferred?

In other words, personalized treatments are specialized but at the same time also increasingly boost the process of acquiring data for each patient. For example, the microscopic characterization of brain tumors in children, which is necessary to plan radiation treatment, consists of multiple magnetic resonance imaging (MRI) acquisitions together with genetic profiling and other clinical indicators as biomarkers. All this information constitutes large data volumes characterizing both the anatomy and the physiology of the tumor and the surrounding organs for an individual before the treatment.

ML and Deep learning (DL) approaches are quite popular and consolidated in other fields, including image processing and large data analytics. Often, DL methods do not need the extraction of physiological and anatomical features that are used to train the network since this process is automatically done by the algorithm. The price to pay is the loss of control in feature management and, consequently, the impossibility of defining the initial clinical hypothesis. Instead, ML approaches need a manual definition of features. Even if this is time-consuming, it allows the formulation of the initial hypothesis and, consequently, the use of established physiological models. This aspect is an advantage in medical research since clinicians and researchers always need to link algorithm output with a biological background.

In summary, single patient-level analysis depends on building data profiles that take shape over time depending on events and therefore become naturally more heterogeneous. Such profiles centered on the patient’s medical history are now routinely organized in electronic health records (EHR). Each patient presents a narrative that appears complicated by the degree of fragmentation of the collected data and the variety of sources from which all evidence types have been obtained.

## 4. Data Integration

### 4.1. Rationale

Data integration aims to combine data from different sources into new valuable information not obtainable from any single source. In general, big data features heterogeneity, scale, and structure specify characteristics that altogether complicate the task of integration. The existing online data integration platforms are either repositories with a defined infrastructure or solutions for deployment [15,16]. The former offers the advantage of fast implementation with limited configuration flexibility (see cBioPortal [17], among others). The latter requires implementations that include data storage, high-performance computing (HPC), and network infrastructure, ensuring fast, secure, and encrypted data transmission.

As oncology relies on a variety of consistent and robust results that need to be reconciled to the benefit of interpretation, the clinical data combined with demographics, multi-dimensional imaging, and genetic sequences typically presents the problem of redundancy to be controlled or reduced considering very different aspects such as data privacy and validation to minimize the risk of treatment failure.

### 4.2. Integrative Analytics

This task refers to how to process data that are stacked together or assembled from various information sources and aims at: modeling the data to predict, make the algorithms learn from new data, and use the new knowledge directly in the model. Big data makes hypothesis-driven approaches less viable and instead suggests a more agnostic strategy by avoiding assumptions and prioritizing insights from data patterns and correlations. A potential problem of this approach is ignoring the mechanism behind data generation, with the risk of underestimating or not considering key physiological aspects [18,19].

In oncology, the retrospective analysis of large patient data sets comprises two phases. The first one aims to identify groups of features or biomarkers that characterize a clinical situation as, for example, the presence of a tumor or its staging, as well as the results of therapy with regression or progression of the disease. The second phase is the validation through independent cohorts of such biomarkers with previously unseen patients. Due to the variability of subjects and conditions, this process elucidates both detection and evolution aspects and leaves labeled data for more accurate reproducibility of research results and reliability of the biomarkers.

### 4.3. Predictive Learning

Predictive learning includes several approaches centered on the identification of patterns, structures, and hierarchies in data. ML techniques have the algorithmic ability of automatic learning from training data sets. Precision and accuracy improve with more available instances, and no initial assumption is usually made for the mechanism. The challenge of properly balancing the accuracy of the results with the run-time of the algorithm is prioritized. Often, the balance can be in favor of one or the other factor, but in a clinical environment, both need to be optimized. A list of algorithms specifically implemented for clinical purposes and already approved by Food and Drug Administration can be found at https://medicalfuturist.com/fda-approved-ai-based-algorithms/ (accessed on 29 December 2022).

Some intelligent algorithms call for further development, especially in specific areas. For instance, anomaly detection identifies rare observations or outliers in the unlabeled data assuming that most of the data examples are normal. Anomaly detection methods look for exceptions and are useful in the detection of rare events [20]. Reinforcement learning (RL) is probably the artificial learning method that better mimics human behavior. When the algorithm decides between two states or classes, it takes action based on the environmental conditions. In the case of positive results, one receives a reward; in the case of failure, a previously gained reward is subtracted [21]. The goal of the algorithm is to maximize the rewards. RL applications in oncology have been proposed, for instance, in the domain of detection of lung nodules [22], and while potentially a tool for supporting clinical decisions, a few challenges have been discussed in precision oncology [23], treatment [24], personalized dose optimization [25] and model-informed precision dosing [26].

In general, developing an accurate predictive model that works on combined data or one that combines predictions is challenging. Being contextual, some data relationships are more robust, consistent, or significant than others which may be more independently obtained and sparsely represented. The same idea applies to predictions, given the underlying data system from which they are obtained, suggesting the need to conceive predictive learning as a multilevel data fusion task in which the structures behind the delivered information should be carefully validated and the performance assessed.

In the medical field, validation of new methods aims to answer two basic questions: (a) Is the application safe? (b) Is it effective? Safety is defined as an acceptable risk of harm to a patient, usually compared to current practice [27]. Determining an acceptable risk involves multiple experts, from scientists to government regulatory bodies. The evaluation of the efficacy deals with the determination of numerical estimators able to quantify the performance of the new method as precision, accuracy, robustness, etc. Despite the importance of the topic, there are no worldwide accepted regulations that define harmonized standards to evaluate the performance of AI methods [28,29,30]. A first group of guidance rules has been defined in the Transparent reporting of a multivariable prediction model for individual prognosis or diagnosis (TRIPOD-AI [31]).

## 5. Image Analytics & Radiomics

### 5.1. Background

Integrative predictive learning has recently found an important contribution from radiomics. The main goal of radiomics [32,33] is to build models with features predictive of clinical outcomes, such as survival and response to therapy, and obtain either imaging biomarkers or integrative biomarkers by combining imaging with clinical, molecular, and genomic information. The features are the building blocks that, once retrieved from the images, can be used to represent the texture of the tumor image and reflect the spatial distribution of the tumor’s intensities or shape. The role of radiomics is, therefore, increasingly crucial in supporting the knowledge about the clinical manifestations of disease and the cellular/subcellular processes that drive them.

For instance, in drug discovery, where a huge chemical space is now available and where AI/ML represents the most versatile tool for research at various stages of drug development [34,35], there is a clear need to add information from radiomics to impact a key domain as the prediction of on- and off-target effects before the compounds are synthesized. The incorporation of radiomics with genomics information, integrated with biochemical properties and target tractability, would sharpen the approaches to target identification, especially in early drug development.

### 5.2. Imaging Biomarkers as the Next Generation of Personalized Drug Development

Imaging biomarkers are adding a new layer to the personalization of medicine, including personalized drug development. Endowed with multiple dimensions, tumors exhibit structural diversity that advanced medical imaging analytics can noninvasively quantify. Radiomics can enable drug developers to discover patterns otherwise not obtainable by simply profiling patients’ tumors across standard procedures. Genetic biomarkers, for instance, support the identification of patients genetically predisposed to certain conditions, but imaging biomarkers can help probabilistic assessments, starting from whether to recommend treatment according to the chance of generating a certain outcome. Together, radiomic and genomic biomarkers can significantly change clinical trial structure and management by improving patient targeting to specific treatments.

The combination of a novel drug’s genomics-supported mechanism of action and phenotypic data obtained by radiomics leads to a superior adaptive clinical trial design regarding the effectiveness of drugs in grouped patients in addressing tumor aggressiveness, metastatic potential, and tumor response to therapy. Radiomics may thus represent an accurate early indicator of treatment effectiveness and accelerate adaptive clinical trials by using imaging data associated with target patient characteristics that significantly show therapy effectiveness early in the trial. As a result, a reduction of the drug development cycle can follow with costs saving associated costs [36,37].

### 5.3. Diagnostic Medical Imaging

Computer Tomography (CT), MRI, UltraSound (USA), and Positron Emission Tomography (PET) are routinely used in clinical practice to detect tumors and determine their stages together with histopathological analysis. With the large volumes of image data that are generated and frequently used for retrospective studies, AI/ML algorithms are widely employed for the detection, lesion characterization, and automatic segmentation of tumors and healthy organs. Despite their good performance, a certain bias is present as the applications are often directed to a limited cohort of patients, making the generalization of methods problematic.

Trials with a consistent number of subjects, more than ten thousand, are usually linked to large screening programs. One example is the evaluation of an AI system for breast cancer screening performed in the UK and USA on a cohort of 28,953 women [38]. In this study, the AI system outperformed all the human readers with an area under curve ROC greater than 11.5%. Moreover, the authors demonstrated that such a system, if used in a double reading process, reduces the workload of the second reader by 88%. Another example is the use of a deep convolutional neural network trained (DCNN) with 42,952 subjects (17,627 patients and 25,325 controls) to detect thyroid cancer on ultrasound images [39]. The DCNN model showed an improved specificity in identifying thyroid cancer patients versus skilled radiologists. On the same trend, DCNNs have been trained with a dataset of 129,450 dermatological clinical images to detect and classify skin cancer against 21 board-certified dermatologists [40]. They demonstrated that the AI model was capable of classifying skin cancer with a level of competence comparable to dermatologists.

### 5.4. Imaging-Guided Therapy

Tumor therapy in cancer patient management can potentially benefit most from AI/ML methods. Many studies aim to solve two main problems: first, find the matching between patient and treatment, and second, predict the prognosis. Many efforts have been directed toward the creation of systems able to extract information from clinical data and generate cancer treatment options. Such platforms, called Decision Support Systems (DSS), are composed of algorithms that continuously learn from imaging, biological, genetic, and health record data and match them with various possible care pathways in order to maximize tumor control and quality of life and minimize toxicity [41]. Platforms DSSs curate.ai, for example, provide a drug combination and an appropriate dosing strategy over time following a phenotypic personalized medicine approach [42]. Similarly, the quadratic phenotypic optimization platform (QPOP) [43] has been used to optimize drug combinations in multiple myeloma [44] and pediatric stem and immune cell therapies [45].

In radiation therapy, DSS platforms are starting to be implemented in conventional Treatment Planning Systems [46]. Their goal is to provide radiation oncologists with a selection of ideal treatment plans to maximize the dose at the target and minimize it in the organ at risk (OARs) with scoring systems. Successful implementation has been reported in the case of Low Dose Rate brachytherapy in the treatment of prostate cancer, where machine learning was able to find the best 3D pattern distribution of radioactive seeds to be implanted [47]. In external beam therapy, the application of AI algorithms aims to automatize the planning process and optimize dosimetric trade-offs [48]. Based on known patient geometry and dose prescription, it is possible to train DNN algorithms to predict dose distribution and generate plans for new patients [49]. This approach has already been implemented in some of the most common tumor locations, including the head and neck [50], nasopharynx [51], lung [52], and rectum [53].

### 5.5. Immunotherapy

Given the established clinical endpoints assessing the value of radiomics in patients undergoing immunotherapy often involves measuring the model performance in stratifying candidate patients, early predicting their response [54], and predicting overall survival (OS) and/or progression-free survival (PFS) time [55]. Among the known limitations of these studies, the use of non-standardized imaging methodologies and different datasets diminishes the potential of outcome radiomics signatures to be externally validated.

As ML methods can identify different features based on performance optimized over selected but limited validation sets, it is expected that an even more challenging validation can be shown with integrative radiomic models analyzing the relationship between radiomic features, immune correlates, tumor mutations, and prognostic scores into radiomics models [56]. However, an interesting direction was recently explored by an imaging biomarker indicating intra-tumoral immune status in NSCLC [57], named the immune ecosystem diversity index. This index was applied in a retrospective study with features extracted using preoperative CT and its prognostic value (overall survival) investigated on patients receiving surgical resection.

Another study [58] applied a 3-class classification random forest algorithm (stable, progression, remission), showing better patient stratification but with performance depending significantly on tumor localization and type, hence identifying three major general critical factors in the training data, the generalization power, and the correlative versus causal relationships that were established. The research guidelines here found involve (a) the standardization of acquisition protocols to target more high-quality patient data and allow multi-center translation studies and (b) the combination of domain- and evidence-driven approaches and hypotheses, (c) the integration of multi-modal (both intra-imaging and multi-omics) data and their temporal dimension (longitudinal data) [59,60,61,62,63,64].

### 5.6. Prognostic Prediction

A substantial number of studies have predicted patient prognosis in regard to local tumor control, toxicity, eventual development of metastases, and estimation of survival [65]. Many DL techniques have been applied to cancer prognosis prediction, with good results in terms of global performance, as in the case of glioblastoma multiforme [66], low-grade glioma [65], mesothelioma, colorectal cancer [67,68] and ovarian cancer [69]. Despite the improved network architecture and algorithms, model precision and accuracy are limited by a few factors.

First, the number of subjects is relatively small compared to the variability of tumor features highly dominated by intrinsic heterogeneity. Second, the samples are often imbalanced in terms of the treatment’s success/failure, making training inefficient and needing data augmentation. Third, sample data are often incomplete or not compatible. This situation, typical of retrospective studies and with a cohort of patients acquired in multi-center clinical studies, forces the examiner to remove patients from the study, limiting further the number of available subjects. Fourth, the integration of high dimensional data, including imaging, genetic sequence, clinical data, and health records, introduces errors, approximations, and inaccuracies. Some solutions have been proposed, such as a complex network approach [70]. Finally, it is difficult to compare model performance among different studies for the lack of standards in algorithm implementation and data collection [71].

In radiomics, interesting developments have involved, for instance, benchmark datasets for independent comparisons and analyses. ImageNet, an image database organized according to the WordNet hierarchy [72], is an example of a benchmark dataset that is frequently used to evaluate CNN models [73,74]. Since another priority is to perform an assessment of radiomic signatures in decentralized multi-center studies in which one node develops the signature and another node independently validates it, an interesting lung prognostic study [75] established an equivalently significant performance for 2-year overall survival by distributed versus centralized validation.

Although encouraging, both reproducibility and generalizability of radiomic signatures remain challenges. Both the extraction of features and the design of models are subject to biases that make a variation in image acquisition, and reconstruction parameters exist in the radiomic workflow and are only partially controllable, with a clear impact on translation [76,77].

## 6. Discussion

The opportunity or potential for further developing radiomics is substantial and involves the oncological field and the non-oncological ones too. This means that highly contextualized salient features will be generated, and their integrability with other types of data will be prioritized when building inference models and customizing radiomic algorithms. Being a non-invasive screening method, radiomics will be very useful in dealing with specific tumors, for instance, brain or head & neck, given the possibility to produce consistent acquisition protocols that facilitate reproducible results.

Radiomics tools that support clinical decision-making are destined to specialize further, as seen before when discussing the relevant or emerging areas of immuno- and radiotherapy. Regarding the assessment of response to the therapy, it is expected that radiomics will contribute more systematically to the process of selecting optimal treatments. Re-assessing phenotypes and biomarkers with the information conveyed by radiomic features and integrated with clinical and omics data will fuel personalized medicine. For instance, refining standardized protocols with the inclusion of clinico-genomic-radiomic signatures obtained from validated prediction models will impact clinical practice.

There are multilevel challenges before a larger consensus on radiomics efficacy can take place and large-scale clinical applicability can be reached:Reprioritizing between retrospective and prospective studies while ensuring the generalizability of results with the inclusion of intra-cohort diversity, among other factors;Building data-harmonized processes, stepping from acquisition and annotation (image space) to mining and pre-processing (quantified data space), and finally covering analytics and post-processing (extracted feature space);Conducting inference based on standardized scalable ML and DL algorithms robust to both sample size and variable imbalance in order to optimize feature selection and significance from computational, radiomic, radiological, and clinical standpoints;Setting benchmark models and expanding the gold standard data sets to allow testing and validation of predictive and prognostic biomarkers with a built-in radiomic signature;Expanding the spatio-temporal dimensions of predictive analyses to include integrated spatial biology/immunology marks with therapy timing and dosage effects accounted by suitable model augmentation and correction.

## 7. Conclusions

The need for better understanding and characterizing problematic diseases like cancer has fueled AI and ML applications to deliver evidence and provide interesting results, although the impacts at a clinical level sometimes remain difficult to assess for a combination of reasons, i.e., technical, methodological, cultural, etc. This is also the scenario that data- and image-driven applications face to consolidate the role of radiomics.

Looking ahead, three interventions are urgent and can be summarized as (a) Validation of methodological approaches sharing a common aspect, the reference to ground-truth (b) Assessment of method precision and accuracy in terms of high reproducibility between similarly characterized and independently selected patient cohorts; (c) Generative incorporation of medical expertise into the radiomic tools to delineate an automated information process ultimately establishing scalable and re-usable clinically significant knowledge.

## Data Availability

All data discussed in the paper were obtained from published sources.

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
