# Peer review of "Translating Data Science Results into Precision Oncology Decisions: A Mini Review"

_jcm, 2023, doi:10.3390/jcm12020438_

Round 1
Reviewer 1 Report
It is an interesting and well-written manuscript. I have several comments/questions:
- Some sentences are a little too long, such as L28-L31 "For example... follow-up" and L375-L378 "Augmenting the... clinical practice". Please reorganise them to make more clear.
- Sections 1.2, 1.3 and 1.4: the authors talked about the generalities of cancer management. It should be better to have a few references.
- L39: "patients and healthy subjects acting possibly as control group": healthy subjects or the persons for whom the cancers were ruled out? They can have begnin pathologies. Please explicit.
- L173-175. Yes indeed it is a real problem, are there in the litterature some tracks to overcome it?
- L179- 181: The second phase is it not the vadidation one? More than "detect either a new tumor or predict its evolution". Please explicit.
- L196 "Reinforcement learning". I would like the authors develop more about it with more references. There is only the ref 15.
- L212 " compared to current practice [17]." I would like to have more references supporting that.
The section on "radiomics", the discussion and the conclusion are suitable for me.
Best regards
Author Response
Thanks for the review. Please find the replies in the file.

Reviewer 2 Report
The authors write a short narrative review on Data Science in oncology focusing on radiomics, AI and ML. Review describes the use of Big data as a predictive prognostic factor and as a guide for systemic therapies and radiotherapy. It is overall well written and god flowing reading.
The discussion paragraph summarizes the text with focus on the challenges of modern oncology.
Author Response
Thanks for the comments. The paper has now a few more references and has been polished.